# Cryo-EM of dynein microtubule-binding domains shows how an axonemal dynein distorts the microtubule

**Samuel E Lacey, Shaoda He, Sjors HW Scheres, Andrew P Carter***

MRC Laboratory of Molecular Biology, Cambridge, United Kingdom

**Abstract** Dyneins are motor proteins responsible for transport in the cytoplasm and the beating of axonemes in cilia and flagella. They bind and release microtubules via a compact microtubule-binding domain (MTBD) at the end of a coiled-coil stalk. We address how cytoplasmic and axonemal dynein MTBDs bind microtubules at near atomic resolution. We decorated microtubules with MTBDs of cytoplasmic dynein-1 and axonemal dynein DNAH7 and determined their cryo-EM structures using helical Relion. The majority of the MTBD is rigid upon binding, with the transition to the high-affinity state controlled by the movement of a single helix at the MTBD interface. DNAH7 contains an 18-residue insertion, found in many axonemal dyneins, that contacts the adjacent protofilament. Unexpectedly, we observe that DNAH7, but not dynein-1, induces large distortions in the microtubule cross-sectional curvature. This raises the possibility that dynein coordination in axonemes is mediated via conformational changes in the microtubule.
DOI: https://doi.org/10.7554/eLife.47145.001

## Introduction

The dynein family is a group of minus-end directed microtubule motors. The two cytoplasmic dyneins (dynein-1 and dynein-2) are involved in long-range movement of cellular cargoes (*Reck-Peterson et al., 2018*; *Roberts et al., 2013*). Multiple inner and outer arm axonemal dyneins power the beating motion in cilia and flagella by sliding adjacent doublet microtubules past each other (*Satir et al., 2014*). All dynein family members share a common architecture, based around a heavy chain that contains a cargo-binding tail region and a force-generating motor domain. The motor consists of a ring of six connected AAA+ subdomains (AAA1-6) with the nucleotide cycle of the first, AAA1, powering movement (*Schmidt and Carter, 2016*). Dyneins bind to microtubules via a small microtubule-binding domain (MTBD) consisting of six short helices (H1-H6) (*Figure 1A*). The MTBD is connected to the AAA +ring by an antiparallel coiled-coil stalk, containing helices CC1 and CC2. It binds to the microtubule at the tubulin intradimer interface (*Carter et al., 2008*).

Nucleotide-dependent conformational changes are transmitted to the MTBD through the stalk to modulate its affinity for microtubules, allowing dynein to bind and release as it steps along the microtubule. The stalk transmits these conformational changes due to its ability to pack into two stable registries (α and β+), representing a half-heptad shift of CC1 relative to CC2 (*Carter et al., 2008*; *Gibbons et al., 2005*; *Kon et al., 2009*; *Kon et al., 2012*; *Schmidt et al., 2015*). The structure of the MTBD in the low microtubule affinity state (stalk in β +registry) has been solved to high-resolution (*Carter et al., 2008*; *Nishikawa et al., 2016*). The relative movement of CC1 towards the ring (stalk in α registry) pulls on the MTBD to create a high-affinity state. The structure of this state was visualised in a landmark 9.7 Å cryo-electron microscopy (cryo-EM) structure of the MTBD bound to microtubules (*Redwine et al., 2012*). A pseudoatomic model was fit into the cryo-EM map using molecular dynamic simulations, and showed large conformational changes throughout the MTBD

**\*For correspondence:**
cartera@mrc-lmb.cam.ac.uk

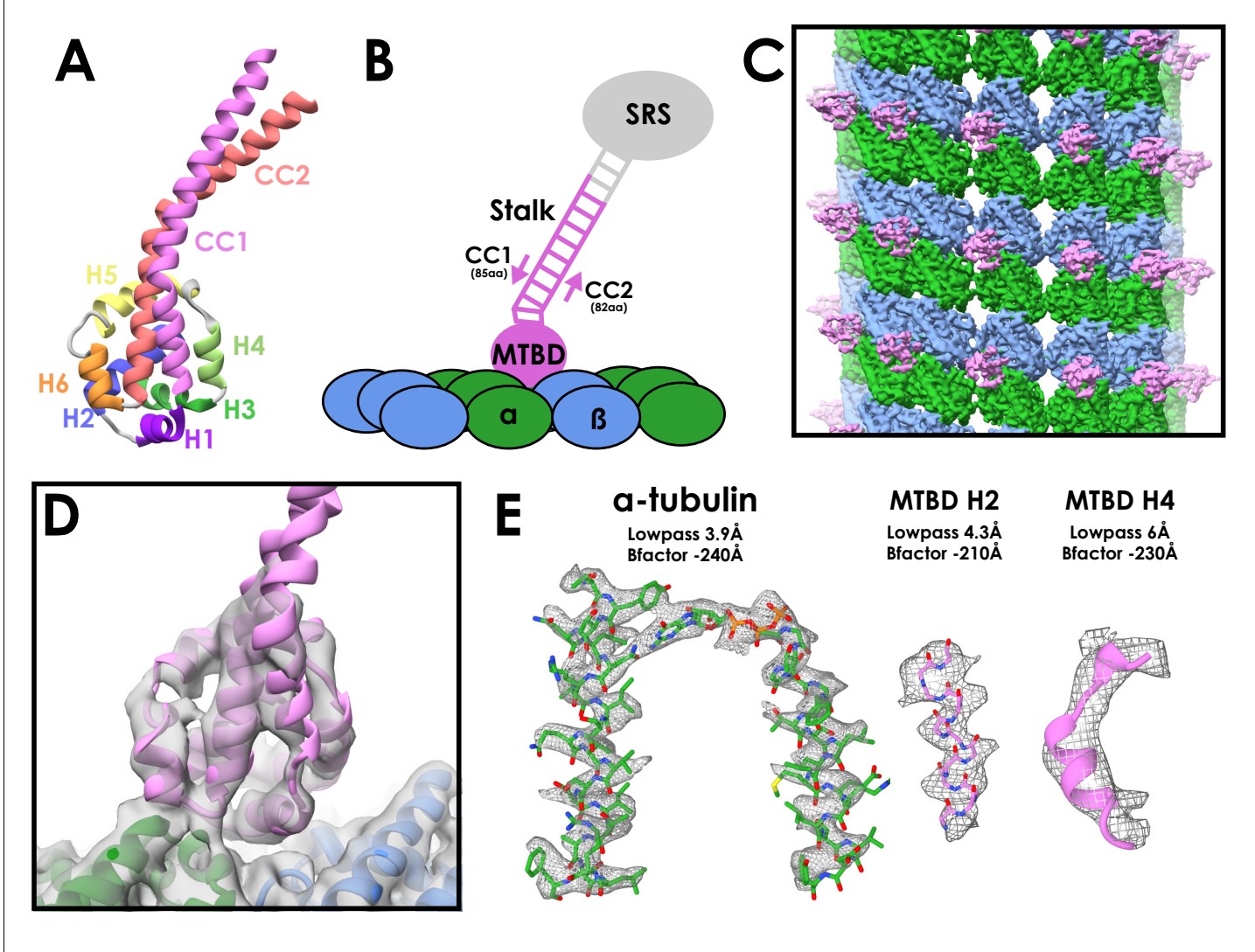

**Figure 1.** Cryo-EM Structure of the cytoplasmic dynein-1 microtubule-binding domain. (**A**) Crystal structure of the cytoplasmic dynein-1 MTBD in the low-affinity β +registry (PDB 3ERR) coloured by helix. (**B**) Schematic of the MTBD constructs used for structure determination. A globular seryl-tRNA synthetase (SRS, grey) has a protruding coiled-coil to which 12-heptads of the dynein stalk is fused (pink). CC1 is three residues longer than CC2 to force the stalk into the high-affinity α registry, allowing the MTBD to bind to the microtubule (α-tubulin in green, β-tubulin in blue) (**C**) Reconstruction of the cytoplasmic dynein-1 MTBD (pink) bound to microtubule (α-tubulin in green, β-tubulin in blue), lowpass-filtered to 5 Å. (**D**) New models for the cytoplasmic dynein MTBD (pink) and tubulin (α-tubulin in green, β-tubulin in blue) was refined into the cryo-EM density (lowpass filtered to 5 Å) (**E**) Representative density of different regions of the map, filtered and sharpened according to local resolution.

DOI: https://doi.org/10.7554/eLife.47145.002

The following figure supplements are available for figure 1:

**Figure supplement 1.** Processing workflow of the SRS-DYNC1H1[3260-3427] structure.
DOI: https://doi.org/10.7554/eLife.47145.003

**Figure supplement 2.** Local resolution of the cytoplasmic dynein-1 MTBD SRS structure.
DOI: https://doi.org/10.7554/eLife.47145.004

**Figure supplement 3.** A comparison between microtubule reconstructions using Relion or previous methods.
DOI: https://doi.org/10.7554/eLife.47145.005

upon binding. The authors concluded that specific interactions between the microtubule and H1, H3, and H6 of the MTBD are required to induce and maintain this high-affinity conformation.

A high degree of conservation in the MTBD between the different dynein family members suggests that this mechanism of microtubule binding is conserved (*Höök and Vallee, 2006*). However,

many axonemal dyneins have an insertion between H2 and H3 called the flap, the function of which is unclear. An NMR structure of the *Chlamydomonas reinhardtii* flagellar dynein-c MTBD showed that the flap consists of two flexible beta-strands extending from the MTBD core (*Kato et al., 2014*). The flap was predicted to sterically clash with the microtubule surface, and therefore undergo rearrangement upon binding.

We decided to take advantage of recent technological advances in cryo-EM (*Fernandez-Leiro and Scheres, 2016*; *He and Scheres, 2017*) to compare the structures of a cytoplasmic and axonemal dynein MTBDs bound to microtubules. We determined the structure of mouse cytoplasmic dynein-1 MTBD on microtubules to an overall resolution of 4.1 Å. We observe a number of structural differences to the 9.7 Å structure. This leads to an updated model for the transition from a low- to high-affinity state, based only around movement of H1 to avoid steric clashes with the microtubule surface. Furthermore, we determine the structure of microtubules decorated with the MTBD of the human inner-arm dynein DNAH7 to 4.5 Å resolution. We show that its flap contacts an adjacent protofilament from the rest of the MTBD, and show that this interaction dramatically distorts the microtubule cross-sectional shape.

## Results

### Structural determination of cytoplasmic dynein-1 MTBD decorating microtubules

To fix the stalk in the high microtubule affinity α registry, we used a chimeric fusion construct (SRS-DYNC1H1$_{3260-3427}$) in which the mouse cytoplasmic dynein-1 MTBD and 12 heptads of stalk are fused to a seryl-tRNA synthetase (*Figure 1B*) (*Carter et al., 2008*). Predominantly 13-protofilament microtubules were made by polymerizing tubulin in an MES-based buffer (*Pierson et al., 1978*). The MTBD and MT were incubated together on-grid and vitreously frozen for cryo-EM (*Figure 1—figure supplement 1A*, *Table 1*).

Microtubules are characterised by having like-for-like lateral contacts (i.e. α-to-α tubulin contacts); however, most microtubule architectures, including 13-protofilament microtubules, break this pattern with a seam of heterotypic α-to-β contacts (*Desai and Mitchison, 1997*) (*Figure 1—figure supplement 1B*). As such, these microtubules cannot be subjected to conventional helical symmetry averaging. Previously, EM structures of microtubules with a seam have used iterative 2D cross-

**Table 1.** Data collection statistics for the presented structures.
* N-fold non-crystallographic symmetry applied.

| | SRS-DYNC1H1$^{3260-3427}$ | DYNC1H1$^{1230-4646}$ | SRS$^{+}$-DNAH7$^{2758-2896}$ |
|---|---|---|---|
| Microscope | Krios | Polar | Krios |
| Detector | Falcon III (Linear) | Falcon III (Linear) | Falcon III (Linear) |
| Voltage (kV) | 300 | 300 | 300 |
| Exposure time (s) | 1.5 | 1.5 | 1.5 |
| Total dose (e-/Å$^2$) | 60 | 67.5 | 55.5 |
| Pixel Size (Å$^2$) | 1.04 | 1.34 | 1.085 |
| Defocus range (μm) | -1.5 to -4.5 | -1.5 to -4.5 | -1.5 to -4.5 |
| Sessions | 1 | 1 | 1 |
| Micrographs | 1995 | 2455 | 5527 |
| Symmetry | 13* | 13* | 9* |
| Final particles | 59100 | 38142 | 41984 |
| Resolution (Å) | 4.1 | 5.5 | 4.5 |

DOI: https://doi.org/10.7554/eLife.47145.006

The following source data is available for Table 1:
**Source data 1.** Data collection statistics for the presented structures.

DOI: https://doi.org/10.7554/eLife.47145.007

correlation to synthetic projections followed by 3D refinements in order to locate the position of the seam (*Sindelar and Downing, 2007*; *Zhang and Nogales, 2015*). We propose an alternative image processing approach that is integrated into the helical Relion pipeline (*Figure 1—figure supplement 1C*, *He and Scheres, 2017*). 3D refinement follows the standard regime for a C1 helix, however 13-fold local, non-point-group symmetry is applied to the reconstruction following each iteration. The application of local symmetry depends on a user-specified mask and symmetry operators that super-impose equivalent tubulin dimers on top of each other. The application of local symmetry increases the signal in the reference during refinement, and prevents progressive deterioration in the definition of the seam from poorly aligned particles. The same local symmetry is also applied to increase the signal in the final reconstruction. Relion has the advantage of requiring minimal user input, and is capable of sorting sample heterogeneity with 2D and 3D classification.

We initially used 2D classification to remove microtubules that were poorly decorated or possess identifiable non-13 protofilament architectures (*Figure 1—figure supplement 1C*). Particles from good 2D classes were used for 3D classification, which resulted in a single good class (*Figure 1—figure supplement 1C*). Manual inspection confirmed that this class contained only 13-protofilament microtubules.

The seam is well defined in the resulting asymmetric reconstruction, with α-tubulins making lateral contact with β-tubulins. This is displayed most clearly when viewing the extended luminal S9-S10 loop in α-tubulin and the additional rise between MTBDs across the seam (*Figure 1—figure supplement 1E/F*). Application of local symmetry resulted in a map with an overall resolution of 4.1 Å (*Figure 1C/D*, *Figure 1—figure supplement 1D*). The resolution of the tubulin density ranges from 3.6 Å to 4.4 Å (*Figure 1E*, *Figure 1—figure supplement 2A*). The tubulin model refined into our density closely matched a previous high-resolution EM model of taxol-stabilised microtubules (*Kellogg et al., 2017*). The density for the dynein MTBD ranges from 4.4 Å to 8 Å in resolution (*Figure 1—figure supplement 2A*). Regions of the MTBD close to the microtubule were of sufficient resolution to see the pitch of alpha helices (*Figure 1E*). We built and refined a new model of the high-affinity form of the MTBD using the low-affinity MTBD crystal structure (*Carter et al., 2008*) as a starting point (*Figure 1D/E*, *Figure 1—figure supplement 2B*, *Table 2*).

## Microtubule binding involves movement of only one helix

Surprisingly, when comparing the low-affinity crystal structure (*Carter et al., 2008*) to our new high-affinity structure, we see that the majority of the MTBD undergoes remarkably few changes upon microtubule binding (*Figure 2A/B*). The Cα RMSD between the two structures for helices H2 to H6 is 1.9 Å. The major movement involves H1 and CC1, which move together to occupy the intra-dimer cleft above α- and β-tubulin. A more minor change is seen in H6 in order to accommodate its interaction with α-tubulin (*Figure 2A*).

This is in contrast to the previous 9.7 Å microtubule-bound model, in which H1 moves towards β-tubulin and shifts H2, H3 and H4 as a result (*Figure 2C/D*, Cα RMSD H2-H6 3.7 Å). We see a much smaller movement in CC1 and H1, and as a result H3 and H4 stay much higher above the microtubule surface. The experimental setup used in this study is slightly different to the previous structure. We imaged a longer 12-heptad stalk-SRS construct on 13 protofilament microtubules, compared to a 3-heptad construct on 14-protofilament microtubules. To confirm that the structure we observed is representative of full-length dynein bound to microtubules, we imaged a human cytoplasmic dynein-1 motor domain construct (DYNC1H1$_{1230-4646}$, *Steinman et al., 2017*) bound to microtubules (*Figure 1—figure supplement 1A*). Grids were frozen in the absence of nucleotide to attain a high-affinity state. Using the Relion pipeline we achieved an overall resolution of 5.5 Å (*Figure 2—figure supplement 1A*, *Figure 2—figure supplement 2A*, *Table 1*). Lower microtubule occupancy was observed for DYNC1H1$_{1230-4646}$ compared to SRS-DYNC1H1$_{3260-3427}$, which is likely a result of steric clashes between adjacent motor domains around the microtubule. Microtubules with high occupancy were selected following 2D classification. The lower occupancy meant that the MTBD density was at a much lower resolution than the microtubule, and therefore the map was lowpass filtered to 8 Å for interpretation. The resulting map fits well with our new model, but H1, H2, H3 and H4 from the 9.7 Å model are all at least partially outside the density (*Figure 2—figure supplement 2C–F*). Accordingly, model-to-map FSC measurements indicate that our new model is a better fit to the map (FSC$_{0.5}$=8.66 Å (New model) and 12.76 Å (9.7 Å model) (*Figure 2—figure supplement 2B*). As

**Table 2.** Model refinement statistics for the presented structures.

| | SRS-DYNC1H1[3260-3427] | SRS+-DNAH7[2758-2896] |
|---|---|---|
| **Map** | | |
| Map resolution (Å) | 5 | 5.4 |
| Map sharpening B factor (Å2) | -150 | -200 |
| Map CC | 0.87 | 0.84 |
| Map:Model FSC$_{0.5}$ (Å) | 5.06 | 5.48 |
| **Model Composition** | | |
| Non-hydrogen atoms | 8001 | 14908 |
| Protein residues | 1001 | 1867 |
| Ligands (GTP/GDP) | 1/1 | 2/2 |
| **R.m.s deviations** | | |
| Bond lengths | 0.009 | 0.01 |
| Bond angles | 1.91 | 1.91 |
| **Validation** | | |
| MolProbity score | 2.35 | 2.49 |
| Clashscore | 9.05 | 9.43 |
| Rotamer Outliers (%) | 5.37 | 6.9 |
| **Ramachandran plot** | | |
| Favored (%) | 95.5 | 94.9 |
| Outliers (%) | 0.3 | 0.65 |
| CB Outliers (%) | 0.53 | 0.98 |

DOI: https://doi.org/10.7554/eLife.47145.008

The following source data is available for Table 2:
**Source data 1.** Model refinement statistics for the presented structures.
DOI: https://doi.org/10.7554/eLife.47145.009

such, we conclude that our 4.1 Å structure corresponds to the native state of the dynein MTBD bound to microtubules.

The MTBD residues forming the interface with the microtubule in our new model are consistent with previous structural and mutagenesis data (*Gibbons et al., 2005*; *Koonce and Tikhonenko, 2000*; *Redwine et al., 2012*). In contrast, some of the residues on the microtubule previously predicted to interact with the MTBD (*Redwine et al., 2012*) are too distant in our model (Red bonds, *Figure 2—figure supplement 2G/H*). For example, K3299 is now over 7 Å away from its previously predicted interaction partner β-tubulin E420, but is directly proximal to D427 (*Figure 2—figure supplement 2G*). Overall, our new model suggests that the MTBD is permanently primed for microtubule binding. The only significant conformational change that occurs upon binding is the movement of H1 upwards to accommodate the change in stalk registry.

## An axonemal dynein MTBD contacts four tubulin subunits at once

We next investigated whether the microtubule interaction we observe is conserved across different dynein families. DNAH7 is a monomeric inner arm axonemal dynein closely related to *Chlamydomonas reinhardti* flagellar dynein-c (*Hom et al., 2011*; *Wickstead and Gull, 2007*). It is one of five human axonemal dyneins to contain a flap insert between H2 and H3 of the MTBD (*Figure 3A* and *Figure 3—figure supplement 1A/B*), and has one of the most divergent MTBD sequences compared to cytoplasmic dynein-1.

A previous NMR study of *Chlamydomonas* flagellar dynein-c reported a low microtubule affinity (*Kato et al., 2014*). In contrast, another study observed that single dynein-c molecules bind well enough to move along microtubules (*Sakakibara et al., 1999*), and an artificial construct fusing the

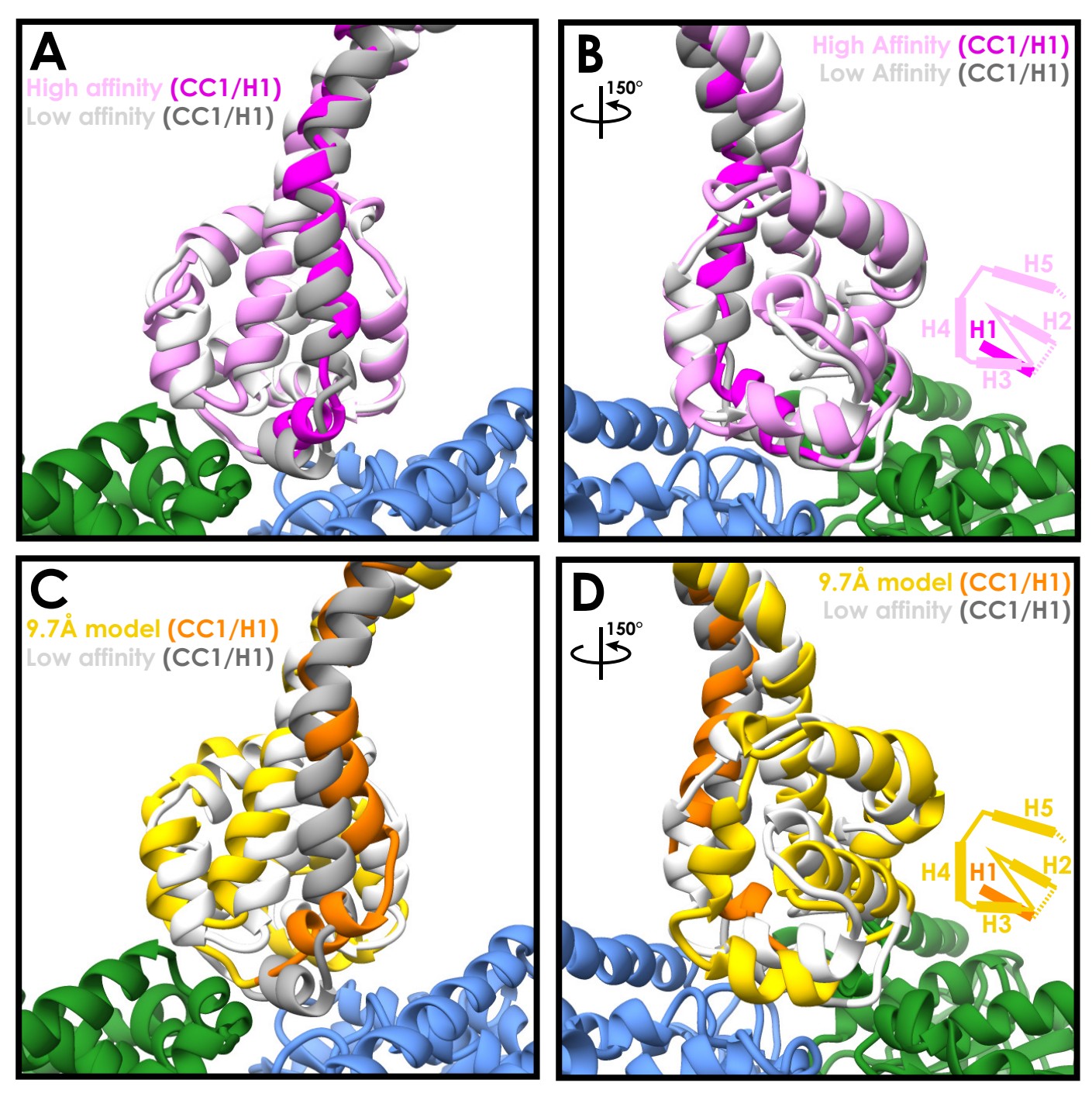

**Figure 2.** Similarities between the high- and low-affinity states of the cytoplasmic dynein-1 MTBD. (A) A comparison between the newly refined cytoplasmic dynein-1 MTBD model (pink) and the low-affinity state crystal structure (PDB 3ERR, docked to the same density, white). CC1 and H1 (highlighted in magenta and grey for high- and low-affinity states respectively) rise in the high-affinity state to solve a steric clash with the microtubule (α-tubulin in green, β-tubulin in blue) (B) Orthogonal view of A, highlighting the similarity between the two models away from CC1/H1. Cartoon displays organisation of H2-H5 as visibile in the model. (C) A comparison the previous 9.7 Å cryo-EM microtubule-bound model (gold, PDB 3J1T) and the low-affinity state crystal structure (white, 3ERR). There is a larger movement up and to the side in CC1 and H1 (orange and grey for 3J1t and 3ERR respectively) in 3J1T compared to the new model. (D) Orthogonal view of C, showing H2, H3 and H4 all in different conformations relative to the microtubule in 9.7 Å model.

DOI: https://doi.org/10.7554/eLife.47145.010

The following figure supplements are available for figure 2:

*Figure 2 continued on next page*

*Figure 2 continued*

**Figure supplement 1.** Processing pipeline for DYNC1H1$_{1230-4646}$ and SRS$^+$-DNAH7 $_{2758-2896}$ structures.
DOI: https://doi.org/10.7554/eLife.47145.011
**Figure supplement 2.** Validation of the SRS-DYNC1H1$^{3260-3427}$ model with the dynein motor domain.
DOI: https://doi.org/10.7554/eLife.47145.012

DNAH7 MTBD onto a cytoplasmic dynein-1 stalk bound to microtubules with high-affinity (*Imai et al., 2015*). We initially expressed and purified a 12-heptad stalk-SRS DNAH7 MTBD construct, but observed poor microtubule decoration in EM. We therefore made an SRS-fusion containing the mouse cytoplasmic dynein-1 stalk and the human DNAH7 MTBD. Consistent with (*Imai et al., 2015*) this construct (SRS$^+$-DNAH7$_{2758-2896}$) exhibited strong decoration of microtubules in cryo-electron micrographs (*Figure 1—figure supplement 1A*). We subjected these decorated microtubules to the same routine of data collection as before (*Table 1*).

Following 3D classification and refinement, we obtained a 4.5 Å resolution map (*Figure 3B*, *Figure 2—figure supplement 1A*, *Figure 3—figure supplement 2A*). A model of the DNAH7 MTBD was refined into the density using a homology model to the low-affinity NMR structure (PDB 2RR7, *Kato et al., 2014*) as a starting point (*Figure 3B*, *Figure 3—figure supplement 2B*, *Table 2*). Comparing the final microtubule-bound model to the original low-affinity structure, we observe the same conformational changes as for SRS-DYNC1H1$_{3260-3427}$ (*Figure 3C*). Namely, the majority of the MTBD remains unchanged, but H1 and CC1 move up into a raised position over the intradimer interface. Furthermore, aligning our cytoplasmic dynein-1 model to the DNAH7 model shows that they adopt almost identical conformations (*Figure 3—figure supplement 2C*).

The biggest difference between the cytoplasmic dynein-1 and DNAH7 models lies in the flap. At lower threshold levels an elongated density emerges from the DNAH7 MTBD and contacts the adjacent protofilament (*Figure 3D* and *Figure 3—figure supplement 2B*). There are two contacts between the flap and this protofilament, corresponding to H10 of β-tubulin and loop H6/7 of α-tubulin (*Figure 3F/G*). As such, DNAH7 contacts four tubulin subunits at once when it binds to the microtubule. On account of its appearance only at lower threshold levels, we conclude that the flap has a degree of flexibility.

Another difference between the two structures is the orientation of the MTBD on the microtubule. Aligning the tubulin of the DNAH7 and cytoplasmic dynein-1 models reveals that the DNAH7 MTBD is tilted relative to cytoplasmic dynein (*Figure 3E*). This can be described by a 7° rotation around the microtubule contact site at the base of H6. DNAH7 is tilted in the same direction as the flap, suggesting that its interaction with the adjacent protofilament pulls on the entire MTBD.

Lowpass filtering and thresholding the map to a lower level revealed the presence of an additional link between the DNAH7 MTDB and the adjacent protofilament (*Figure 3H*). Due to its proximity to the C-terminus of β-tubulin, we suspect that this density is attributed to the β-tubulin 'E-hook'. This is a normally unstructured ~20 residue chain of mostly glutamate residues that strongly contributes to the electronegative surface of microtubules (*Nogales et al., 1998*; *Redeker et al., 1992*). We note that the same link is not observed in our cytoplasmic dynein-1 density (*Figure 3I*) at any threshold. Examination of the surface charges of our two models reveals a large positively charged patch on the top of the DNAH7 MTBD that is absent in cytoplasmic dynein-1 (*Figure 3J/K*). Modelling of the β-tubulin E-hook show that it is capable of reaching this patch. As such, we conclude that on top of a core microtubule interaction shared with cytoplasmic dynein, DNAH7 is bolstered by two links to the adjacent protofilament.

## DNAH7 binding to microtubules induces changes in microtubule cross section

During the process of solving the DNAH7 MTBD structure, we noticed that the microtubule was distorted compared to our other reconstructions. Initial application of local symmetry, using the same operators as our previous structures, resulted in a weaker decorating density and blurring of tubulin helices (*Figure 3—figure supplement 2D/E*). This suggested that the averaging between different protofilaments was incoherent, and the cross-section of the microtubule was distorted. We therefore used Relion to refine the local symmetry operators, which improved the features and led to our final 4.5 Å map (*Figure 3—figure supplement 2F*).

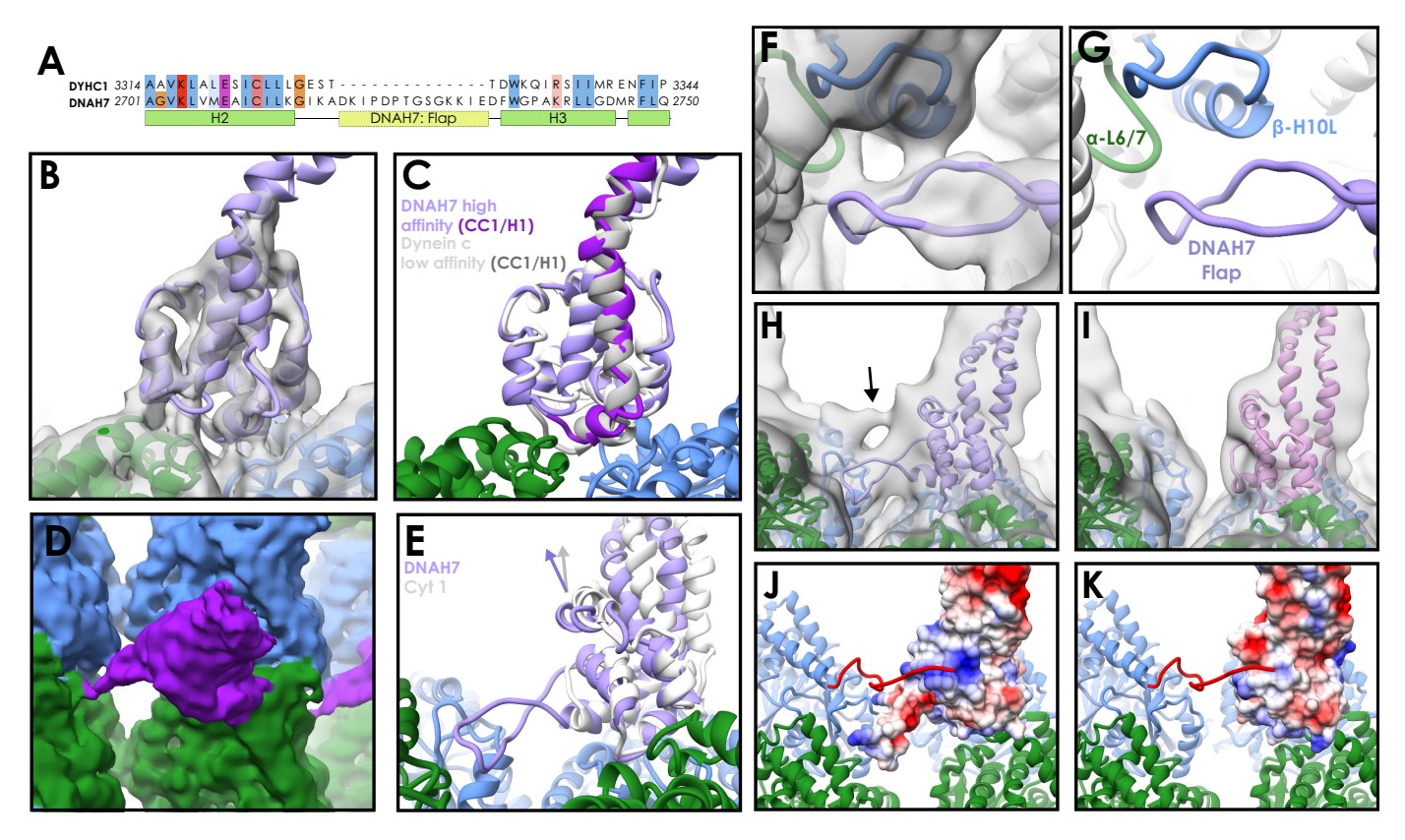

**Figure 3.** Structure of the DNAH7 MTBD. (A) Partial sequence alignment of human cytoplasmic dynein-1 (DYHC1) and human Axonemal Dynein 7 (DNAH7). DNAH7 has a 13-residue insert between H2 and H3 compared to cytoplasmic dynein called the flap. Full MTBD sequence alignment in *Figure 3—figure supplement 1* (B) A model of DNAH7 (violet) was refined into the SRS$^+$-DNAH7$_{2758-2896}$ cryo-EM density (grey, lowpass filtered to 5 Å). (C) A comparison between the microtubule-bound DNAH7 model (violet) and the low-affinity Flagellar dynein C NMR structure (*Kato et al., 2014*, 2RR7, ensemble chain 8, docked to the same density, white). As with cytoplasmic dynein, the only major conformational changes between the two is the upward movement of CC1/H1 (purple and grey for DNAH7 and 2RR7 respectively) (D) The SRS$^+$-DNAH7$_{2758-2896}$ cryo-EM density (MTBD in purple, α-tubulin in green, β-tubulin in blue) thresholded at a low level, revealing the extension of the flap to contact the adjacent protofilament (E) Rotation of the DNAH7 MTBD relative to cytoplasmic dynein-1. The tubulin was aligned in the cytoplasmic and axonemal dynein models. This reveals a rotation around the base of H6/CC2 towards the adjacent protofilament in DNAH7 (violet) compared to cytoplasmic dynein-1 (white). (F) Close-up view of the flap of the DNAH7 MTBD model with its corresponding density. (G) Corresponding view to F, without the electron density. (H) The DNAH7 map lowpass filtered to 10 Å reveals a new connecting density that cannot be explained by the flap (DNAH7 model docked, violet). (I) The extra density observed in H is not seen in the cytoplasmic dynein-1 density filtered to the same resolution (J) The C-terminal tail of β-tubulin could potentially explain the density in H, given its length and possible binding site in a positively charged pocket on the top surface of DNAH7 (surface coloured by coulombic potential) (K) Cytoplasmic dynein-1 (surface coloured by coulombic potential) does not have the same positive patch on DNAH7, suggesting that the C-terminal tail would not dock in the same way.

DOI: https://doi.org/10.7554/eLife.47145.013

The following figure supplements are available for figure 3:

**Figure supplement 1.** Sequence alignments of dynein microtubule-binding domains.
DOI: https://doi.org/10.7554/eLife.47145.014

**Figure supplement 2.** Assessment of the DNAH7 MTBD structure.
DOI: https://doi.org/10.7554/eLife.47145.015

To explore the distortion, we manually measured the rotation between each protofilament relative to the long axis of the microtubule (*Figure 4A*). Measurements for the SRS-DYNC1H1$_{3260-3427}$ map were as expected, with the angle between each protofilament close to 27.69° (360°/13 protofilaments, S.D. = 0.17°) (*Figure 4B*). In contrast, our refined SRS$^+$-DNAH7$_{2758-2896}$ structure exhibited protofilament angles ranging from 26.5° to 28.7° (S.D = 0.75°, *Figure 4B*), representing large changes in the local curvature of the microtubule.

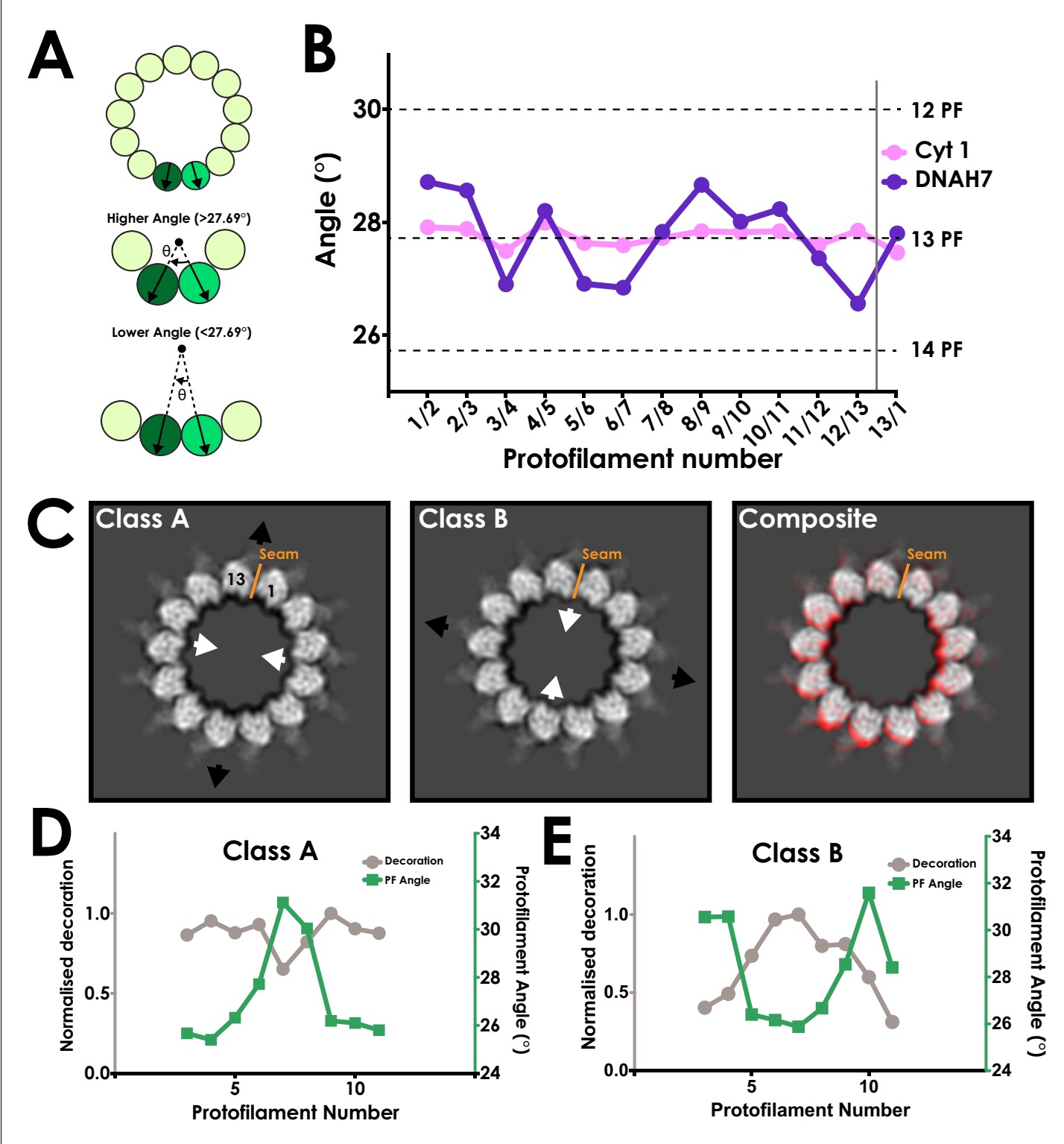

**Figure 4.** DNAH7 binding causes cross-sectional deformations in the microtubule. (**A**) Schematic representing measurement of protofilament angles. A PDB model for tubulin (PDB 5SYF) was docked into adjacent protofilaments, and the rotation to superimpose the two models was measured. Local curvature between each neighbouring protofilament can thus be measured to match a canonical 13-protofilament microtubule (27.69° rotation), or be more or less curved. (**B**) A plot of the angle between each neighbouring pair of protofilaments in the cytoplasmic dynein-1 (Cyt 1, pink) and DNAH7 (purple) maps prior to symmetrization. In DNAH7, the microtubule has been distorted from a perfect circle. Protofilament angles from canonical 12-PF (30°), 13-PF (27.69°) and 14-PF (25.71°) are shown, and the position of the seam is indicated by the grey line (**C**) A projection of one helical rise of Class A

*Figure 4 continued on next page*

*Figure 4 continued*

and Class B (left and center) with arrows indicating distortion in the curvature of the microtubule. Note that Class B and Class C are squashed and extended in reciprocal directions. (Right) A composite image showing the distortion between Class A (Red) and Class BC (white) (**D**) A plot of the protofilament angles (green) and relative decoration (grey) on each protofilament for class A. (**E**) A plot of the protofilament angles (green) and relative decoration (grey) on each protofilament for class B.

DOI: https://doi.org/10.7554/eLife.47145.016

The following figure supplement is available for figure 4:

**Figure supplement 1.** Decoration levels in classes A and B.

DOI: https://doi.org/10.7554/eLife.47145.017

Our initial 3D classification of the SRS$^+$-DNAH7$_{2758-2896}$ particles resulted in a single class used in the refinement. However, to investigate the curvature further, we reclassified the SRS$^+$-DNAH7$_{2758-2896}$ data with modified classification parameters (see Materials and methods, *Figure 3—figure supplement 2H*). This resulted in two extreme classes, 'A' and 'B', with large differences in cross-sectional curvature (*Figure 4C*, *Video 1*). The formation of multiple good classes indicates that the microtubules in this dataset form a continuous distribution of distorted curvatures. Classes A and B had even greater local cross-sectional curvature distortions than our refined map, with protofilament angles ranging from 25.1° to 31.8° (*Figure 4D/E*). The local curvatures between protofilaments that result from these distortions are thus within the range of those seen in canonical 12 (25.71°) and 14 (30°) protofilament microtubules.

These distortions change the microtubule cross-section from a circular to an elliptical profile. The ellipticity (the ratio between the long and short diameters of an ellipse) of class A and B is 0.936 and 0.942, respectively (*Figure 3—figure supplement 2I/J*). This is much greater than microtubule ellipticity measured both in our cytoplasmic dynein-1 structure (0.995, *Figure 3—figure supplement 2K*) and previous studies (*Kellogg et al., 2017*), indicating that DNAH7 is responsible for this distortion.

To determine how DNAH7 binding affects the curvature of the microtubule, we measured the relationship between the local curvature of the microtubule and the level of MTBD decoration in classes A and B. We observe a clear correlation in which the decoration is highest at low local protofilament curvatures (*Figure 4D/E*, *Figure 4—figure supplement 1*).

To investigate the relationship between decoration level and local curvature at the level of individual particles, we performed sub-classification of our 4.5 Å DNAH7 dataset. We made masks encompassing a single MTBD and the two tubulin dimers it contacts, subtracted the signal outside the mask and performed 3D classification on the resulting particles. We repeated the analysis for particles taken from the pairs of protofilaments with the highest and lowest local curvature (*Figure 5A*).

Local curvature within sub-classes ranged from 23.6° to 32.5° (*Figure 5B*), highlighting the structural heterogeneity within DNAH7 decorated microtubules. We measured the level of MTBD decoration for each sub-class relative to the sub-class with the highest decoration and plotted these values against local curvature (*Figure 5C*). This shows a linear relationship between decoration level and local curvature, with the lowest curvature (flattest) sub-classes having the highest decoration. In all the sub-classes, we see evidence of the flap contacting the adjacent protofilament, suggesting that the main differences in curvature result from the degree of decoration.

Based on our observations, we propose that the DNAH7 MTBD induces a local flattening of the microtubule. The pattern of decoration suggests that the DNAH7 MTBD binds more weakly

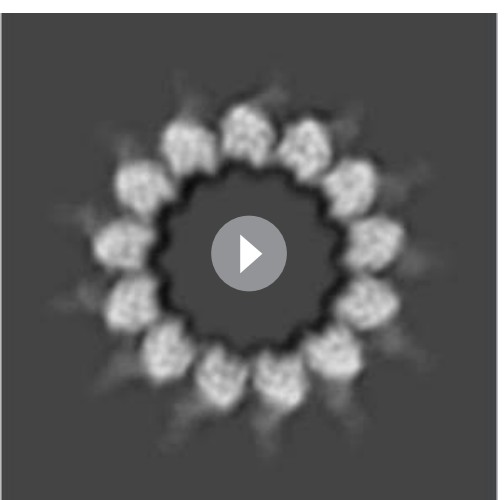

**Video 1.** A morph between Classes A and B, depicting their cross-sectional distortion.

DOI: https://doi.org/10.7554/eLife.47145.018

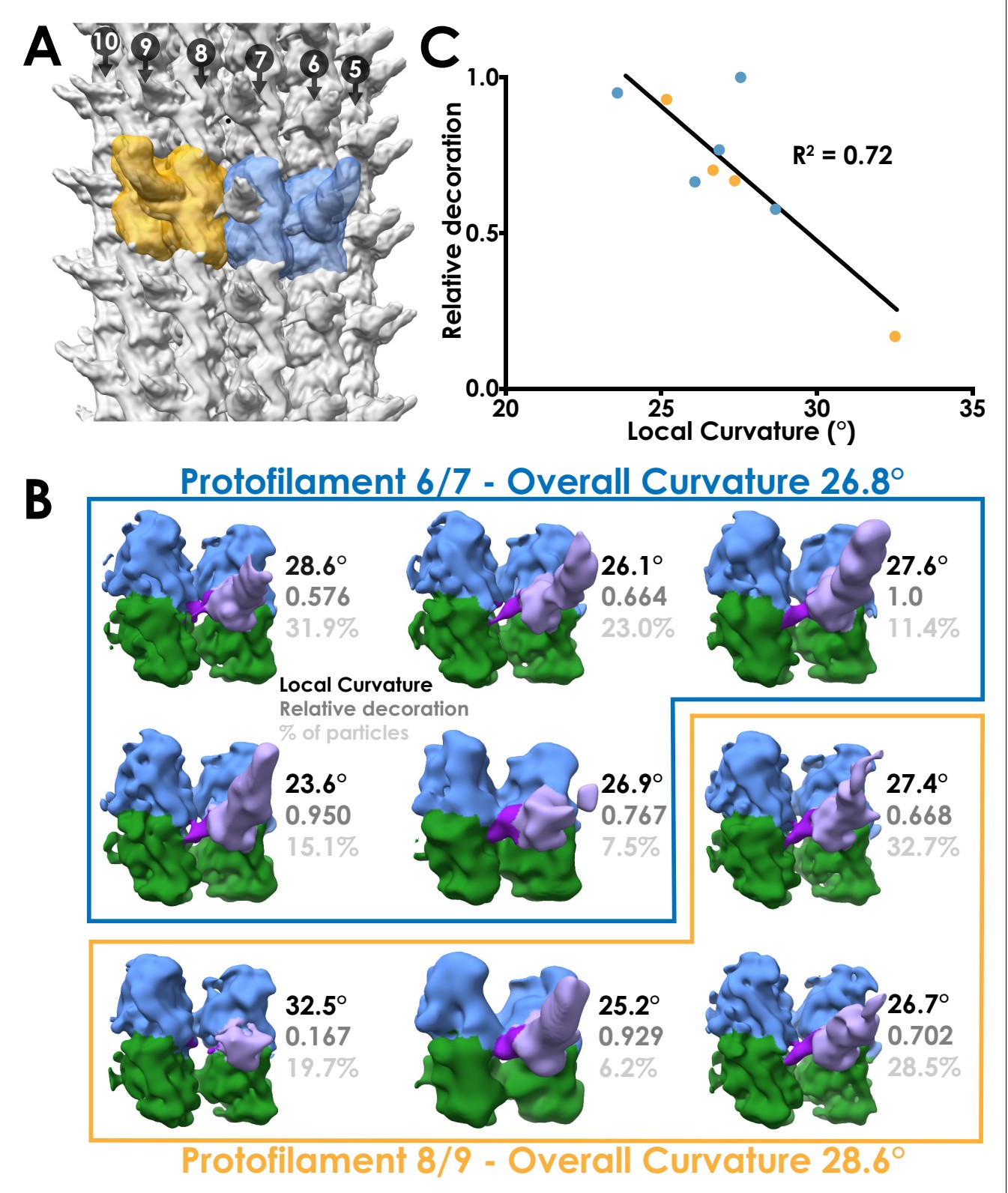

**Figure 5.** Local sub-classification of DNAH7 decorated microtubules. (**A**) The two masks used for focused sub-classification of the DNAH7 dataset, encompassing one MTDD and two tubulin dimers (blue/orange). One mask (blue, protofilaments 6/7) corresponded to the region with the lowest overall curvature (26.8°, *Figure 4B*) and the other (orange, protofilaments 8/9) corresponded to the region with the highest overall curvature (28.6°, *Figure 4B*). (**B**) The well populated sub-classes from 3D sub-classification of the DNAH7 dataset (MTBD/flap violet/purple, α-tubulin green, β-tubulin

*Figure 5 continued on next page*

*Figure 5 continued*

blue). Sub-classes are in two independent groups from the low or high curvature protofilaments (blue/orange boxes). Local curvature (black), decoration relative to the most highly decorated sub-class (mid-grey) and percentage of particles contributing to the sub-class from the same job (light-grey) indicated. (C) A plot of the relationship between local curvature between tubulin dimers within a sub-class and decoration level. Line from linear regression analysis ($R^2$ = 0.72, N = 9, Y = −0.0863*X + 3.058).

DOI: https://doi.org/10.7554/eLife.47145.019

to higher curvatures and more strongly to flatter curvatures.

Given that the only major difference between the cytoplasmic dynein-1 and DNAH7 MTBD is the flap, it seems likely that this is the element responsible for the flattening. This is supported by our observation that the flap contacts the adjacent protofilament to the MTBDs binding site. In our DNAH7 MTBD sub-classes (*Figure 5B*), we do not see any evidence of docked MTBDs without flap binding, suggesting that the flap contact is an integral feature of the DNAH7 microtubule interaction. Furthermore, the movement of the MTBD towards the adjacent protofilament suggests that there is tension created by the flap. We propose that this tension results in the relative movement between protofilaments that generates a local flattening.

## Discussion

### Using Relion to solve high-resolution structures of decorated pseudosymmetric microtubules

The workflow we present for solving microtubule structures in Relion is straightforward and does not require expert knowledge. To test the general applicability for decorated microtubules, we re-processed EMPIAR dataset 10030, which comprises EB3-decorated microtubules. This data set was previously refined to 3.5 Å resolution using an iterative seam-finding protocol (*Zhang and Nogales, 2015*; *Zhang and Nogales, 2015*). Following the Relion pipeline with the same data resulted in an asymmetric map with lower levels of density on the protofilaments either side of the seam, suggesting that not all particles were aligned correctly with respect to the seam (*Figure 1—figure supplement 3D/E*). However, after symmetrization the map extends to 3.5 Å resolution and is essentially identical in appearance to the published structure (*Figure 1—figure supplement 3A–C*). We conclude that Relion can be used to reconstruct pseudosymmetric microtubules and their decorating partners to high resolution.

We note that many of the previously introduced methods for microtubule reconstruction explicitly determine, and enforce, a single seam orientation for each microtubule (*Sindelar and Downing, 2007*; *Zhang and Nogales, 2015*). In Relion, the in-plane rotation, translations, and out-of-plane rocking (tilt) angle can be restrained to similar values in neighbouring segments from each microtubule through the use of Gaussian priors. However, such a prior has not been implemented for the first Euler angle, that is the rotation around the helical axis. Therefore, each segment is aligned independently from its neighbours in the same microtubule, and mistakes in alignment of the seam can be made on a per-segment basis. Therefore, at least in its current implementation, Relion appears to require a larger decorating density (such as the dynein MTBD) to fully align the seam.

### A revised high-affinity state of the dynein MTBD

Our new 4.1 Å reconstruction of the cytoplasmic dynein-1 MTBD allows us to identify the structural transitions that occur upon dynein binding to microtubules. The MTBD interface made up of H2, H3, H4 and H6 shows only minor changes upon microtubule binding, suggesting it is in a binding-primed conformation regardless of nucleotide state. This interface can initiate contact with the microtubule, but a stable microtubule-bound state cannot be achieved when dynein is in a low-affinity state due to a steric clash between H1 and β-tubulin. When the motor switches to a high microtubule affinity nucleotide state, H1 moves up and the MTBD can be further stabilised on the microtubule by interactions between H1 and β-tubulin. Conversely, when ATP binds to the motor, changes in the stalk (*Schmidt et al., 2015*) push H1 down and release the MTBD from the microtubule.

Combining our current work with previous studies (*Carter et al., 2008*; *Kato et al., 2014*), we now have structures of two distantly related dynein MTBDs both on and off the microtubule. In both

the cytoplasmic dynein-1 and DNAH7 MTBDs, the predominant change is the position of H1, suggesting this is a conserved mechanism for controlling dynein binding to microtubules. However, the addition of an extended flap between H2 and H3 in DNAH7 results in important differences to cytoplasmic dynein. The flap extends the contact area with the microtubule to include four tubulin subunits, supplementing the core interaction observed in cytoplasmic dynein-1. We propose that this interaction drives large distortions in lattice curvature. We assign a second link to the adjacent protofilament to the acidic C-terminal tail of beta-tubulin binding to a pocket of positively charged residues on the top surface of the DNAH7 MTBD. The angle the stalk of DNAH7 makes relative to the microtubule is tilted in an off-axis direction compared to cytoplasmic dynein-1. It is interesting to note that many inner arm dyneins generate torque (*Vale and Toyoshima, 1988*; *Kagami and Kamiya R, 1992*; *Kikushima and Kamiya, 2008*) and we speculate that the tilt may be one mechanism that could contribute to this.

## DNAH7-induced distortions of the microtubule

Some members of the kinesin family of microtubule motors are known to induce changes in the microtubule lattice. These include a longitudinal extension by kinesin-1 (*Peet et al., 2018*) and the peeling back of single protofilaments by kinesin-13 (*Hunter et al., 2003*). Both these effects are related to the longitudinal axis however, and no effects on cross-sectional curvature have been observed to date. In contrast, some microtubule-associated proteins are sensitive to the lateral curvature of the microtubule. Doublecortin (DCX) and EB3 both bind in the cleft between two adjacent protofilaments and contact four tubulin subunits (*Moores et al., 2004*; *Zhang and Nogales, 2015*). These lateral interactions promote formation of 13-protofilament microtubules during polymerization. This promotes a rounder, more regular lattice. Therefore, DNAH7 is unusual in directly distorting the cross-sectional curvature of mature microtubules.

A number of other axonemal dyneins possess an MTBD flap (*Figure 3—figure supplement 1B/C*). In *C. reinhardtii*, the γ outer arm dynein and the a, b, c, d and e inner arm dyneins all possess an 18/19-residue loop between H2 and H3 (*Figure 3—figure supplement 1C*). Cryo-electron tomography has mapped the position of each dynein MTBD in the axoneme (*Figure 6A*) (*Bui et al., 2008*; *Liu et al., 2008*; *Nicastro et al., 2006*; *Song et al., 2018*). Microtubules in the axoneme form a doublet structure, consisting of a 13-protofilament A-tubule and a connected 10-protofilament B-tubule. The tail of axonemal dyneins dock onto the A-tubule, positioning the MTBDs to bind to an adjacent doublet B-tubule. Strikingly, the flap containing dyneins are spread along the length of the axonemal repeat but only contact two pairs of protofilaments on one side of the B-tubule (*Figure 6B*). As such, there is one side of the B-tubule on which flap-containing dyneins act at a high local concentration. This is in contrast to our DNAH7 decorated microtubule structure, in which near-saturation binding around the entire microtubule may confound local effects of individual DNAH7 MTBD binding. The effect of dynein binding to microtubules in the axoneme may therefore result in even greater distortions than those we observed. Conversely, as our study was performed on in vitro polymerized microtubules we cannot rule out that DNAH7 binding has a different effect on axonemal doublets, where for example inner proteins may increase the microtubule stiffness (*Ichikawa et al., 2017*; *Ichikawa et al., 2019*).

Axonemal dyneins drive ciliary bending (*King, 2018*; *Satir et al., 2014*). Most models suggest that localised activation and inhibition of axonemal dyneins is needed to create the imbalance in forces across the axoneme that result in an overall bend. These models rely on communication between axonemal dyneins such that they are active or inactive at the right time. We speculate that the flap could play a role in communication. We observed higher decoration at lower lateral curvature in our classes (*Figure 4D/E*), suggesting that the DNAH7 MTBD could act as a curvature sensor in the axoneme. We note that the inner proteins maintain the local curvature of the inner arm dynein binding site at ~25°, which would favour DNAH7 binding (*Carter et al., 2008*). The cross-section of microtubules is thought to flatten during bending (*Memet et al., 2018*), in which case dynein binding would cyclically change during the axonemal beat. Alternatively, the flap-induced distortion of the microtubule could induce cooperative binding of adjacent dyneins, potentially helping a waveform spread through the cilium. Higher resolution structures of axonemal dyneins in the context of a ciliary beat will be required to test these hypotheses.

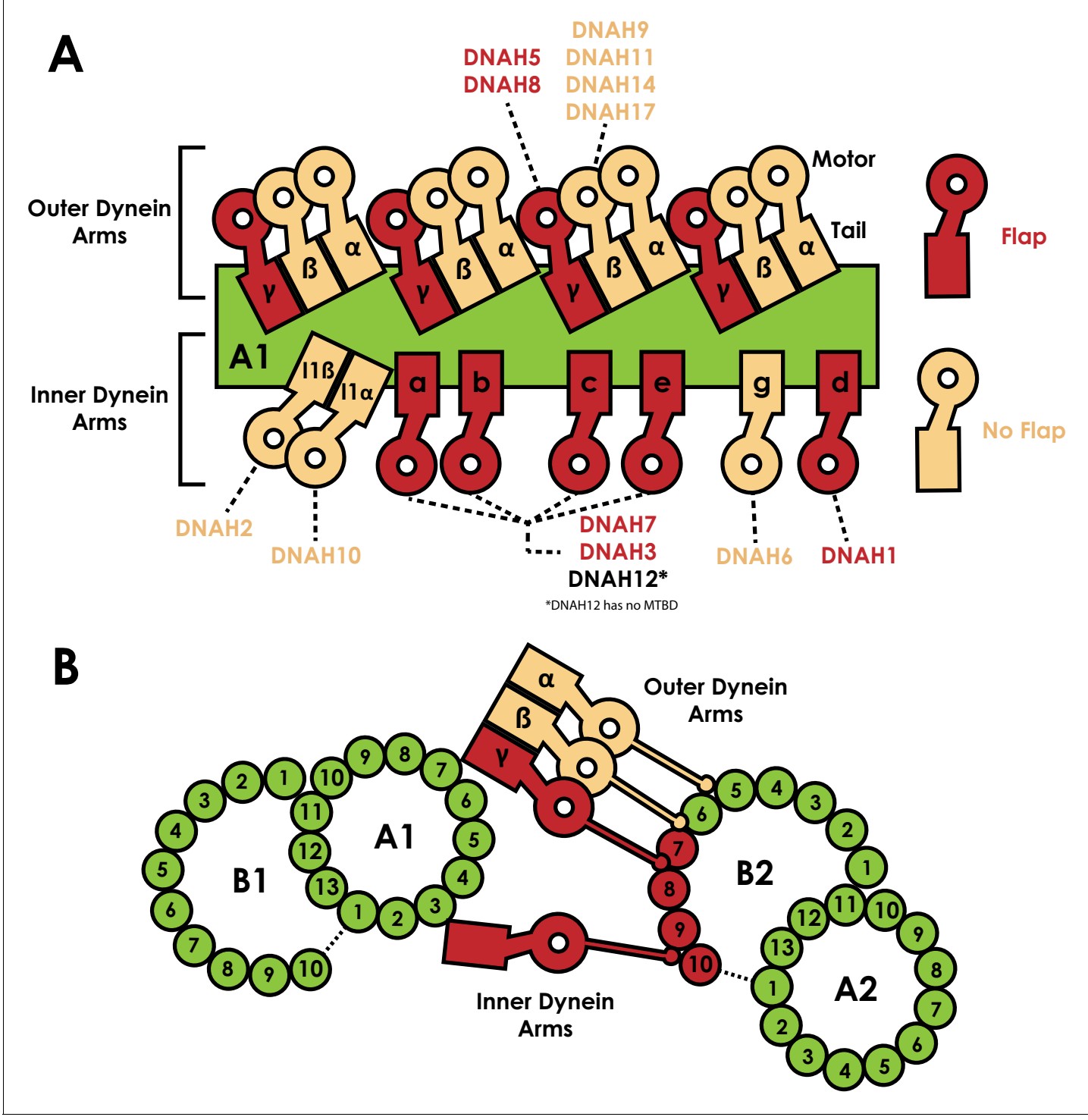

**Figure 6.** Distribution of dyneins in the axoneme. (**A**) Schematic view of the positions the dynein heavy chain tails dock on the A-tubule of the microtubule doublet (green) in the 96 nm axonemal repeat, based on previous structural data on *C. reinhardtii* and Sea urchin sperm flagella (*Bui et al., 2008*; *Lin and Nicastro, 2018*; *Nicastro et al., 2006*). Heavy chains are named according to the *C. reinhardtii* nomenclature, and the human orthologues are linked. Heavy chains are colour coded based on whether they possess a MTBD flap (Red) or not (beige). (**B**) Orthogonal view of A, now looking down the flagella. Links between the inner and outer dynein arms and the exact protofilament of the adjacent doublet has also been established by structural work (*Song et al., 2018*). The position of the tails on tubule A1 positions the MTBDs on tubule B2. We note each of protofilaments 7 to 10 on tubule B2 are contacted by flap-containing dynein MTBDs.

DOI: https://doi.org/10.7554/eLife.47145.020

## Materials and methods

### Protein preparations

6xHis 12-heptad SRS fusion proteins were expressed in SoluBL21 *Escherichia coli* cells (Invitrogen) from a pet42a vector. The mouse SRS-DYNC1H1$_{3260-3427}$ construct is identical to the SRS-MTBD-85:82 used in *Carter et al. (2008)*. The DNAH7 MTBD and stalk was made as a synthetic gene product (EpochGene). SRS$^+$-DNAH7$_{2758-2896}$ was made by cloning the DNAH7 sequence into SRS-DYNC1H1$_{3260-3427}$ in the place of the MTBD, delineated by the universally conserved proline residues as in *Imai et al. (2015)*. Cells were grown in LB media at 37°C until their OD$_{600}$ measured 0.4–0.6, at which point they were supplemented with 1 mM IPTG and grown for 16 hr at 16°C. Cultures were spun at 4000x rcf for 15 min, and used directly for purification. Both SRS constructs were purified according to the same protocol.

A 1L pellet was resuspended in 50 mL Lysis buffer (50 mM Tris pH8.0, 100 mM NaCl, 1 mM MgCl$_2$, 10% Glycerol, 10 mM Imidazole pH8.0, 1 mM DTT, 2 mM PMSF) and lysed by sonication. The lysate was centrifuged at 30,000x rcf in a Ti70 rotor (Beckman) for 30 min and at 4°C. The supernatant was loaded onto a 5 mL NiNTA HisTrap HP Column (GE), washed with 10 column volumes of 10% elution buffer (Lysis buffer with 500 mM Imidazole pH 8.0 and without PMSF) and eluted with a step gradient to 40% elution buffer. Peak fractions were pooled and concentrated in a 15 mL 30kMWCO centrifugal concentrator (Amicon) to a concentration of ~5 mg/mL. Aliquots were snap frozen in liquid nitrogen.

ZZ-tagged human cytoplasmic dynein one motor domain (DYNC1H1$_{1230-4646}$) was cloned into pFastBac and expressed in *Sf9* insect cells as in *Schmidt et al. (2015)*. A 1L pellet was resuspended in 50 mL ZZ-Lysis buffer (as above but without imidazole) and dounce homogenised with 30 strokes. Lysate was centrifuged at 50,0000x rcf in a Ti70 rotor (Beckman) for 60 min and at 4°C. Supernatant was mixed with 2 mL IgG Sepharose 6 Fast Flow resin (GE, equilibrated in ZZ-lysis buffer) on a horizontal roller for 2 hr at 4°C. The mixture was applied to a gravity flow column, and the resin was washed with 150 mL ZZ-Lysis buffer and 150 mL TEV buffer (50 mM Tris pH 7.4, 150 mM KOAc, 2 mM MgAc, 1 mM EGTA, 10% Glycerol, 1 mM DTT). The resin was resuspended in 5 mL TEV buffer, supplemented with 0.1 mg/mL TEV protease and incubated on a horizontal roller at 25°C for 80 min. The sample was reapplied to a gravity flow column, the eluate was collected and concentrated to 6 mg/mL with a 15 mL 100kMWCO centrifugal concentrator (Amicon) and snap frozen in aliquots.

Aliquots of each sample were gel filtered prior to each grid freezing session. Thawed sample was spun through a 0.22 um spin filter (Amicon) to remove aggregates and loaded onto a Superose 6 10/300 gel filtration column (GE) equilibrated in GF buffer (25 mM Tris pH8.0, 50 mM NaCl, 1 mM MgCl$_2$, 1 mM DTT). Peak fractions were pooled and concentrated in a 4 mL 30MWCO Amicon centrifugal concentrator to 1/10$^{th}$ of the original volume. The sample was then diluted fivefold in salt-free GF buffer (i.e. without 50 mM NaCl) and reconcentrated. This was repeated twice, resulting in a 25-fold dilution of the NaCl. The sample was further diluted to a final concentration of 2 mg/mL to be used for grid freezing.

Lyophilised tubulin was resuspended in MES-NaCl buffer (25 mM MES pH6.5, 70 mM NaCl, 1 mM MgCl$_2$, 1 mM DTT) to a concentration of 10 mg/mL and snap frozen in aliquots. For polymerisation, an aliquot was thawed and mixed 1:1 with MES-NaCl buffer supplemented with 6 mM GTP, and incubated at 37°C for 2 hr. 100 µL MES-NaCl buffer supplemented with 20 µM Taxol and prewarmed to 37°C was added, and the sample was left at room temperature overnight. Before use, the microtubules were spun at 20,000x rcf for 10 min, and resuspended in MES-NaCl buffer with taxol.

### Grid preparation

Quantifoil R1.2/1.3 Au300 grids were glow-discharged for 40 s. 4 µL 0.4 mg/mL microtubules was added to the grid and incubated at room temperature for 1 min. This was removed by side blotting, 4 µL of dynein was added and the grid was incubated for a further 2 min. Manual side blotting was repeated, and after the second MTBD application the grid was taken into the humidity chamber of a Vitrobot Mark II set to 100% humidity and 22°C. After 2 min, the grid was double-side blotted for 4 s and plunged into liquid ethane.

## Cryo-electron microscopy

Cytoplasmic dynein 1 MTBD-SRS grids were imaged on our in-house Titan Krios microscope, and DNAH7 MTBD grids were imaged on Krios III at Diamond eBIC. For cytoplasmic dynein, 1995 1.5 s exposures were collected with a pixel size of 1.04 $\text{Å}^2$ and a flux of 40e⁻/$\text{Å}^2$s on a Falcon III detector in linear mode. For DNAH7, 4641 1.5 s exposures were collected with a pixel size of 1.085 $\text{Å}^2$ and a flux of 45e⁻/$\text{Å}^2$s. Dynein motor domain decorated microtubules were imaged on a Polara microscope, with 2455 1.5 s exposures collected with a pixel size of 1.34 $\text{Å}^2$ and a flux of 37e⁻/$\text{Å}^2$s on a Falcon III detector in linear mode. In each case, images were acquired with a defocus ranging between −1.5 μm and −4.5 μm semi-automatically in EPU.

## Image processing

All processings were performed inside the Relion 3.0 pipeline (*Zivanov et al., 2018*). Details are given for processing the SRS-DYNC1H1$_{3260-3427}$ data, followed by modifications to this workflow used for the other datasets. The unaligned raw movies were aligned and dose weighted in Relion's implementation of MotionCorr2 using 4 × 4 patches (*Zheng et al., 2017*). CTF determination was performed with Gctf on dose-weighted micrographs (*Zhang, 2016*). Manual picking and 2D classification was performed to generate references for autopicking. Start and end coordinates of 30 microtubules from five micrographs were extracted into 82 Å segments (box size 512), resulting in ~650 particles which were classified into five classes. These were used as references for autopicking on all the micrographs, using the following parameters: mask diameter 497 Å, in-plane angular sampling 1°, lowpass references 20 Å, picking threshold 0.04, minimum inter-particle distance 79 Å, maximum stddev noise 1.4, shrink factor 0.5, helical picking, tube diameter 400 Å, helical rise 82 Å, number of asymmetrical units 1, maximum curvature 0.4. Particles were extracted (4x binned) and entered for 2D classification into 100 classes. Classes were rejected if they were obviously not microtubules (carbon, ice etc), if they appeared blurred or poorly aligned, if they had low levels of decoration and if they showed signs of non-13-PF architectures (*Figure 1—figure supplement 1C*). A 3D reference was made by docking a model of dynein MTBD decorated tubulin into density for a 13-PF microtubule (PDB 3J1T and EMD 6351 respectively). The PDB was converted to electron density in EMAN2 (*pdb2mrc*). 3D classification of unbinned particles into three classes was used to separate out the remaining sample heterogeneity. The single good class was entered into a 3D refinement using the following parameters: initial angular sampling of 0.9°, and initial offset range and step sizes of 3 and 1 pixels, respectively. C1 symmetry, inner tube diameter 100 Å, outer tube diameter 400 Å, angular search range tilt 15°, psi 10°, tilt prior fixed, range factor of local averaging 4, helical symmetry with one asymmetric unit, initial rise 82 Å, initial twist 0°, central Z length 40%, local searches of symmetry, rise search 78–86 Å, step size 1 Å, no search for twist.

A solvent mask and 13-fold local symmetry were applied during refinement. For local symmetry, a mask was made by docking copies of PDB 3J1T into the protofilament to the left of the seam (if the MT is being viewed plus-end up). The PDB protofilament was then converted to electron density with the EMAN program *pdb2mrc*. This was converted into a mask with *relion_mask_create*. *relion_local_symmetry* requires a STAR file containing the translational and rotational operators needed to move the original mask onto each successive protofilament. The psi angle, rotating around the microtubule long axis, is given as multiples of −27.69° (360°/13). The centre of rotation is the centre of the microtubule lumen, so the only translation needed is the rise between adjacent protofilaments. For a three-start helix, there is a rise of 1.5 dimers through 360°. The refined helical rise between dimers in the same protofilament as measured by Relion was 82.29 Å. As such, the _rlnOriginZ parameter increases by multiples of 9.495 Å (82.293 * 1.5/13). Local symmetry was applied during refinement with the additional argument –local_symmetry. Following completion of refinement, local symmetry was applied to both unfiltered half maps. Postprocessing and resolution assessment was performed with three tubulin dimers docked along a single protofilament as previously (*Zhang et al., 2015*, *Zhang et al., 2018*).

Refinement of the dynein motor domain also followed this protocol. For the DNAH7 structure, initial 3D classification did not result in a coherent class. Instead all good particles following 2D classification were entered into 3D refinement, resulting in a map with blurred features. Following this, 3D classification into eight classes using the orientations used in the refinement (i.e. with no image alignment) was performed. Local symmetry operators were found for the resulting map with the

search command in *relion_localsym* (see https://www2.mrc-lmb.cam.ac.uk/relion/index.php?title=Local_symmetry). The seam was less well defined in the DNAH7 structure, presumably due to local curvature being a stronger feature than seam position in some particles during refinement. As a result, three seam-adjacent profilaments were not included for symmetrisation.

For DNAH7 MTBD sub-classification, signal subtraction was performed on the refined DNAH7 structure. Using the refined DNAH7 model to make a mask, everything but one MTBD and the two tubulin dimers it contacts was subtracted from the raw particles. This was performed for two protofilament pairs (6/7 and 8/9). These particles were subjected to masked 3D classification without alignments (15 classes, T = 100, 25 iterations, limit resolution E-step to 15 Å). For classes A and B, another 3D classification (eight classes, T = 20, 25 iterations, 0.9° local angular searches) was performed with the original unsubtracted particles.

The EB3 dataset was downloaded from EMPIAR (ID 10030) and processed as for the dynein MTBD with modifications. 3D classification was skipped since the microtubules in this dataset almost exclusively have 13 protofilaments (*Zhang and Nogales, 2015*). EB3 does not bind across the seam, which means that applying regular 13-fold symmetry was not appropriate. A separate mask was created for the tubulin and EB3 densities. The tubulin mask and EB3 masks were applied with 13- and 12-fold symmetry, respectively.

Local resolution estimation was performed in *relion_postprocess*.

## Model building

For cytoplasmic dynein 1 a low-affinity crystal structure (PDB 3ERR) was used as the starting model for refinement. For human DNAH7, a sequence alignment to *C. reinhardtii* flagellar dynein c was used to generate a homology model to the low-affinity NMR structure (PDB 2RR7) in Modeller (*Sali and Blundell, 1993*). The homology model was used as an initial model. The models were fit in their respective maps using Chimera (*Pettersen et al., 2004*). A tubulin dimer was also docked in to the density using PDB 5SYF (for SRS$^+$-DNAH7$_{2758-2896}$ the tubulin dimer being contacted by the flap was added as well). Coot Real Space refine zone (*Emsley and Cowtan, 2004*) was used to manually fit the model to the density, followed by whole model refinement using Refmac5 in the CCP-EM suite (*Brown et al., 2015*; *Burnley et al., 2017*). These two steps were performed iteratively until the model to map measures were maximised. For model to map FSC curves (*Figure 2—figure supplement 2B*), phenix.mtriage was used (*Adams et al., 2010*). All model visualisations were performed in Chimera (*Moores et al., 2004*).

## Analysis of microtubule distortion

Protofilament angles were measured by docking a tubulin dimer pdb model (5SYF) into two adjacent protofilaments of the relevant reconstruction in Chimera. The relative rotation was measured with the 'measure rotation' command. Ellipticity was measured with the Matlab fit_ellipse script deposited in the Mathworks file exchange (https://www.mathworks.com/matlabcentral/fileexchange/3215-fit_ellipse). x,y coordinates for each protofilament were obtained from maximum intensity projections of each class, binarised with the same threshold in FIJI (*Schindelin et al., 2012*). The centre of mass of each protofilament was used as the coordinate. Decoration level was determined by the volume of the zoned MTBD density in chimera at set thresholds.

## Acknowledgements

This study was supported by the MRC-LMB EM facility. We thank C Sindelar for initial help with microtubule processing; C Savva, G Cannone, J Grimmett, and T Darling for help and technical support at the MRC-LMB; S Neumann and J Gilchrist for technical support at Diamond; F Abid Ali and C Lau for comments on the manuscript. We acknowledge Diamond for access and support of the Cryo-EM facilities at the UK national electron bio-imaging centre (eBIC), (proposal EM17434), funded by the Wellcome Trust, MRC and BBSRC. This work was funded by grants from the Wellcome Trust (WT210711) and the Medical Research Council, UK (MC_UP_A025_1011) to APC and the Medical Research Council, UK (MC_UP_A025_1011) to SHWS.

## Additional information

### Competing interests
Andrew P Carter: Reviewing editor, *eLife*. Sjors HW Scheres: Reviewing editor, *eLife*. The other authors declare that no competing interests exist.

### Funding

| Funder | Grant reference number | Author |
|---|---|---|
| Wellcome Trust | WT210711 | Andrew P Carter |
| Medical Research Council | MC_UP_A025_1011 | Sjors HW Scheres<br>Andrew P Carter |

The funders had no role in study design, data collection and interpretation, or the decision to submit the work for publication.

### Author contributions
Samuel E Lacey, Conceptualization, Data curation, Formal analysis, Validation, Investigation, Visualization, Methodology, Writing—original draft, Writing—review and editing, Prepared samples, collected data and performed all cryo-EM processing and analysis, Wrote manuscript with APC; Shaoda He, Software, Methodology, Implemented non-crystallographic symmetry in Relion and provided advice; Sjors HW Scheres, Software, Writing—review and editing, Implemented non-crystallographic symmetry in Relion and provided advice, Provided input to manuscript; Andrew P Carter, Conceptualization, Supervision, Funding acquisition, Writing—review and editing, Supervised project, Wrote manuscript with SEL

### Author ORCIDs
Samuel E Lacey (iD) https://orcid.org/0000-0002-3807-8888
Sjors HW Scheres (iD) https://orcid.org/0000-0002-0462-6540
Andrew P Carter (iD) https://orcid.org/0000-0001-7292-5430

### Decision letter and Author response
Decision letter https://doi.org/10.7554/eLife.47145.033
Author response https://doi.org/10.7554/eLife.47145.034

## Additional files

### Supplementary files
• Transparent reporting form
DOI: https://doi.org/10.7554/eLife.47145.021

### Data availability
CryoEM maps and PDB models have been released in the EMDB and PDB respectively.

The following datasets were generated:

| Author(s) | Year | Dataset title | Dataset URL | Database and Identifier |
|---|---|---|---|---|
| Lacey SE, He S, Scheres SHW, Carter AP | 2019 | CryoEM maps | http://www.ebi.ac.uk/pdbe/entry/emdb/EMD-10060 | EMDataBank, EMD-10060 |
| Lacey SE, He S, Scheres SHW, Carter AP | 2019 | CryoEM maps | http://www.ebi.ac.uk/pdbe/entry/emdb/EMD-10061 | EMDataBank, EMD-10061 |
| Lacey SE, He S, Scheres SHW, Carter AP | 2019 | PDB models | http://www.rcsb.org/structure/6RZA | Protein Data Bank, 6RZA |

| Lacey SE, He S, Scheres SHW, Carter AP | 2019 | PDB models | http://www.rcsb.org/structure/6RZB | Protein Data Bank, 6RZB |

The following previously published dataset was used:

| Author(s) | Year | Dataset title | Dataset URL | Database and Identifier |
|---|---|---|---|---|
| Zhang R, Nogales E | 2015 | EMPIAR dataset of EB3 decorated MTs | https://www.ebi.ac.uk/pdbe/emdb/empiar/entry/10030/ | EMPIAR, 10030 |

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
