## [Decision Letter]

Thank you for submitting your article "Cryo-EM of dynein microtubule-binding domains shows how an axonemal dynein distorts the microtubule" for consideration by *eLife*. Your article has been reviewed by three peer reviewers, one of whom is a member of our Board of Reviewing Editors, and the evaluation has been overseen by a Reviewing Editor and Anna Akhmanova as the Senior Editor. The following individual involved in review of your submission have agreed to reveal his identity: Richard J McKenney (Reviewer #3).

The reviewers have discussed the reviews with one another and the Reviewing Editor has drafted this decision to help you prepare a revised submission.

From the Reviewing Editor:

All three reviewers have easily reached a consensus on your work. We would like to accept it for publication after just minor changes are made. While a few experiments were mentioned in our reviews, we do not feel that any of them are required for publication. We ask only that you address all comments that can be addressed in the text. In particular toning down statements related to (1) the universality of the mechanism of microtubule binding because the axonemal dynein was linked to a cytoplasmic dynein stalk, and (2) the deformations of the microtubule because these experiments were not done on axonemal microtubules. All other suggestions or questions that can be dealt with in the text should also be addressed. I have included the full reviews from each reviewer below.

*Reviewer #1:*

Cytoplasmic dynein-1 traffics organelles and other cargos in the cytoplasm, while axonemal dyneins generate movement of flagella. Lacey et al. report the near-atomic resolution structures of the microtubule binding domain (MTBD) of cytoplasmic dynein-1 and an axonemal dynein, DNAH7. They solved both structures using new cryo-electron microscopy analysis methods implemented in Relion. These methods should be simpler for other labs to use compared to previous methods for solving helical structures. There are two main finding from this work. First, an updated model of cytoplasmic dynein-1's MTBD is presented. A previous structure obtained by combining lower resolution cryo-EM and molecular dynamics showed multiple rearrangements in the MTBD upon contact with the microtubule. Of these, only the movement of helix 1 is seen in the new high-resolution structure. Importantly, Lacey et al. report the helix 1 movement in both a construct containing a short region of anti-parallel coiled coil linked to the MTBD and in the context of a construct comprising the entire motor domain of cytoplasmic dynein-1. Second, they present the first high-resolution structure of an axonemal dynein MTBD bound to microtubules. This structure showed a similar overall mechanism of microtubule binding, a movement of helix 1, when compared to cytoplasmic dynein-1. An extension, or "flap", that had been observed in a solution structure of another axonemal dynein was also observed and found to interact with the neighboring microtubule protofilament. In what is arguably the most interesting finding in this paper, the authors observe distortions in the microtubule when the DNAH7 MTBD is bound to microtubules and discuss how this might relate to flagellar beating.

Essential revisions:

1) DNAH7 has low microtubule binding affinity with its own coiled-coil stalk, but high microtubule binding affinity when linked to the coiled-coil stalk of cytoplasmic dynein-1. There are at least two possible explanations for this. First, the dynein-1 stalk is better at stabilizing the high affinity state of the MTBD without a qualitative change in the MTBD's native conformations. Second, there could be more significant changes induced by the dynein-1 stalk compared to a native DNAH7 stalk, leading to artificially similar structures. Absent additional data, the authors should tone down the statement about universal conservation of the mechanism of microtubule binding.

2) For Figure 2B and D, I found it confusing that H1 wasn't represented in the stick model. It could be shown behind H3 in the darker pink color. The statement in the legend "as visible in the model" could be misinterpreted as meaning that other elements were not seen in the structures.

3) For Figure 3D, it would help the reader if each α and β tubulin density were colored in the two tones of green and labeled α and β.

4) In Figure 5A, why are the labels for DNA9, 11, and 17 in orange?

*Reviewer #2:*

In this manuscript, Lacey and his colleagues analyzed 3D structure of cytoplasmic and axonemal dynein microtubule-binding domains (MTBDs) binding to the microtubule, using cryo-EM technique. They carefully analyzed complex structure of MT-MTBD at unprecedented high resolution and updated the previous knowledge. They also proved that their new high resolution structure corresponds to the native dynein binding to MT.

They examined alternative binding of axonemal dynein, compared to cytoplasmic dynein, which proved that MTBD of axonemal dynein has contact to four tubulins (cytoplasmic dynein has only two contacts) and overall angle of binding is ~7degrees rotated. Moreover, they found a clue of flexible C-terminal of β-tubulin reoriented and fixed upon binding of axonemal dynein.

Another interesting finding is that binding of axonemal dynein MTBD induces distortion of MT lattice structure, making it rather elliptic instead of cylindrical.

Overall the new facts they provided by their high resolution cryo-EM study are landmark to characterize this important motor protein and will activate further research in this field. The results are presented beautifully in the graphical figures and methods are explained clearly. Considering the wealth of their findings, this manuscript deserves publication in any high journal and this reviewer enthusiastically recommends it for *eLife* after revising of the manuscript under the following viewpoints.

Title: Flap binding to the adjacent PF is an excellent discovery in this work. This reviewer would like to have it in the title, while the current title mentions only its influence on the MT lattice structure. Maybe "Axonemal dynein MTBD binds to 4 tubulins and distort MT structures" or "High resolution cryo-EM structure of axonemal dynein and MT revealed molecular mechanism of MTBD binding and its influence on MT lattice" will be suitable.

Subsection “Structural determination of cytoplasmic dynein-1 MTBD decorating microtubules”: This reviewer could not understand this sentence and cited figure panels. What "extension" does it mean? Does it mean that the tubulin core stretches? Or does it indicate any extended structure (like S9-S10 loop mentioned in Figure 1—figure supplement 1E) (if so, please indicate it in the Figure 1—figure supplement 2A)? Or is it phase extension to improve resolution – in this case it should read "from 4.4A to 3.6A resolution"? Please clarify this.

CTF panels such as Figure 1D should be improved. The letters in the footnote are too small and are hard to read.

Dynein c generates torque during in vitro motility assay, according to Oiwa's work, while another dynein (probably dynein f) is now to drive MT straightly. How is it interpreted to be linked with the unique interface between DNAH7 and tubulin?

Does 3D classification (of axonemal dynein and its MTBD, especially) show any sign of structural heterogeneity, for example detached flap from the adjacent PF?

Their finding may suggest that the energy landscape is more stable with elliptic MT upon binding of axonemal dyneins. Elliptic shape of B-tubule was reported in Maheshwari et al., (2015) Structure and Ichikawa et al., (2017). Is the orientation of MT flattening caused by dynein-binding consistent with the flattening of B-tubule?

*Reviewer #3:*

The manuscript by Lacey et al., reports high-resolution cryo-EM structures of the microtubule binding domains from cytoplasmic and axonemal dynein motors bound to microtubules. In order to accomplish this, the authors have developed novel image processing routines to aid in solving the structures of proteins bound to the microtubule lattice, providing a new tool for the structural biology community. While the structures of these dynein MTBDs have been examined previously, this is the first study to elucidate their structures bound to microtubules. The authors find substantial differences in the structural movements of the cytoplasmic dynein MTBD from that reported previously in Redwine et al., a study that relied on much lower resolution cryo structures and molecular dynamics simulations. Thus, this study provides an updated model for how the dynein MTBD binds MTs during the motor's mechanochemical cycle.

Maybe more interestingly, the authors find that an insert in the axonemal MTBD makes a distinct contact with the neighboring protofilament in the MT, and also observe a likely interaction between the B-tubulin C-terminal tail and an acidic patch on the dynein MTBD. They go on to show that the binding of this MTBD distorts the MT lattice, possibly thorough this alternative binding interface via the MTBD flap insert. They show that the axonemal MTBD prefers to bind to MTs at regions of low protofilament curvature, possibly providing an explanation for how axonemal dyneins sense curvature during flagellar beating. The final observation is very interesting; that the axonemal dyneins containing the flap insert are all closely spaced within the axonemal repeat structure, and thus all of them make contact with only a small subset of protofilaments on the B2 doublet.

Overall, I think this is an outstanding paper, of very high interest in both biological insight and technological advance. I am not qualified to review the details of their cryo-EM algorithms, so I will take the author's word for the quality of the new technique. I find the biological insights to be stimulating, and think they open up very interesting new questions about the role of distinct axonemal dynein classes in flagellar movement.

The only concern I have is that the MT distortion is observed on reconstituted MTs made from brain tubulin, an unnatural lattice for the axonemal MTBD. Since the natural lattice for this domain is the B1 doublet, how can we be sure the distortion observed is not an artifact of using single MTs? The authors should probably mention this caveat. Additionally, the authors make a fairly convincing argument that the flap binding to a neighboring protofilament is what may cause the lattice distortion. A formal test of this hypothesis would make the manuscript stronger. Mutation or deletion of the flap region could be done to examine if this restores the MT lattice to its original shape.

---

## [Author Response]

Reviewer #1:[…] Essential revisions:1) DNAH7 has low microtubule binding affinity with its own coiled-coil stalk, but high microtubule binding affinity when linked to the coiled-coil stalk of cytoplasmic dynein-1. There are at least two possible explanations for this. First, the dynein-1 stalk is better at stabilizing the high affinity state of the MTBD without a qualitative change in the MTBD's native conformations. Second, there could be more significant changes induced by the dynein-1 stalk compared to a native DNAH7 stalk, leading to artificially similar structures. Absent additional data, the authors should tone down the statement about universal conservation of the mechanism of microtubule binding.

In order to tone down the discussion, we have removed line from subsection “An axonemal dynein MTBD contacts 4 tubulin subunits at once” reading “Given that these are two distantly related dyneins, this suggests that the fundamental structural basis for microtubule binding is universal”. We have changed the title of subsection“A revised high-affinity state of the dynein MTBD” (from “A universal high-affinity…”). We have also changed the senctence to read “the predominant change is the position of H1, suggesting this is a conserved mechanism…” rather than “universal mechanism” in subsection “An axonemal dynein MTBD contacts 4 tubulin subunits at once”

2) For Figure 2B and D, I found it confusing that H1 wasn't represented in the stick model. It could be shown behind H3 in the darker pink color. The statement in the legend "as visible in the model" could be misinterpreted as meaning that other elements were not seen in the structures.

We have changed the schematic in Figure 2B and 2D to include the position of helix 1 as suggested.

3) For Figure 3D, it would help the reader if each α and β tubulin density were colored in the two tones of green and labeled α and β.

As suggested, we have added the colour of the tubulin density to Figure 3D.

Related to this, we have changed the colour scheme of α and β tubulin from dark and light green to green and blue respectively following external feedback from our BioRxiv submission. We think this makes the differentiation between the two subunits clearer. This has been applied to all figures containing tubulin models. To prevent colour duplications in the same figure, we have changed the 9.7Å high-affinity MTBD model from blue to gold, In Figure 2C/D and Figure 2—figure supplement 2E/F.

4) In Figure 5A, why are the labels for DNA9, 11, and 17 in orange?

This was a mistake that we have now rectified.

Reviewer #2:

[…] Considering the wealth of their findings, this manuscript deserves publication in any high journal and this reviewer enthusiastically recommends it for eLife after revising of the manuscript under the following viewpoints.Title: Flap binding to the adjacent PF is an excellent discovery in this work. This reviewer would like to have it in the title, while the current title mentions only its influence on the MT lattice structure. Maybe "Axonemal dynein MTBD binds to 4 tubulins and distort MT structures" or "High resolution cryo-EM structure of axonemal dynein and MT revealed molecular mechanism of MTBD binding and its influence on MT lattice" will be suitable.

We have considered a number of variations on the reviewer’s suggested titles. In the end we would like to stick with our original title as it highlights the fact that the paper contains both cytoplasmic and axonemal dynein MTBD structures.

Subsection “Structural determination of cytoplasmic dynein-1 MTBD decorating microtubules”: This reviewer could not understand this sentence and cited figure panels. What "extension" does it mean? Does it mean that the tubulin core stretches? Or does it indicate any extended structure (like S9-S10 loop mentioned in Figure 1—figure supplement 1E) (if so, please indicate it in the Figure 1—figure supplement 2A)? Or is it phase extension to improve resolution – in this case it should read "from 4.4A to 3.6A resolution"? Please clarify this.

We were aiming to describe the resolution and quality of the structure in the tubulin regions. We have reworded the sentence to read “The resolution of the tubulin density ranges from 3.6Å to 4.4Å”.

CTF panels such as Figure 1D should be improved. The letters in the footnote are too small and are hard to read.

We have reformatted both the axis legends and the footnote key to make the text larger and more legible. These changes have been applied to Figure 1—figure supplement 1D, Figure 1—figure supplement 3A, Figure 2—figure supplement 2A and Figure 3—figure supplement 2A.

*Dynein c generates torque during* in vitro *motility assay, according to Oiwa's work, while another dynein (probably dynein f) is now to drive MT straightly. How is it interpreted to be linked with the unique interface between DNAH7 and tubulin?*

The reviewer raises an interesting point regarding a key characteristic of axonemal dyneins. In *Chlamydomonas,* dynein a, c, d, e and g all rotate microtubules in vitro. This is thought to fine-tune the ciliary beat into a specific defined waveform.

It was recently shown that the direction in which the linker swings relative to the microtubule controls dynein’s trajectory (Can et al., 2019). For example, an engineered dynein in which the linker swing is ~10° off-axis results in helical motility with a pitch angle of ~10°. It is possible that the ~7° tilt of the DNAH7 MTBD on the microtubule could orient the linker swing off-axis and lead to torque generation. However, it is also possible that other mechanisms are responsible, such as the presence of a proline residue in CC1 which may create a bend in the stalk (Hirose, 2012). We briefly add a note reflecting this to the Discussion section.

Does 3D classification (of axonemal dynein and its MTBD, especially) show any sign of structural heterogeneity, for example detached flap from the adjacent PF?

To address this point, we performed 3D sub-classification on the DNAH7 MTBD to look for heterogeneity. To do this, we used signal subtraction to remove everything except the MTBD and the two tubulin dimers it contacts. To investigate variability in different regions of the structure, we did this once for the most curved pair of protofilaments and once for the least curved pair. We then performed 3D classification without alignments on each of the two regions.

The flap was always visible in decorated classes, and generally appeared to form the same conformation. However, we were able to measure the local curvature and decoration intensity in each of our classes. When plotting these two variables against each other, we see a strong correlation between high decoration and low curvature angles. We have included these results as a new Figure 5, with accompanying text in subsection “DNAH7 binding to microtubules induces changes in microtubule cross section”. We feel that these results provide further evidence for the link between DNAH7 binding and microtubule flattening, and thus strengthen the arguments made in the manuscript. The pre-existing Figure 5 is now Figure 6.

Their finding may suggest that the energy landscape is more stable with elliptic MT upon binding of axonemal dyneins. Elliptic shape of B-tubule was reported in Maheshwari et al., (2015) Structure and Ichikawa et al., (2017). Is the orientation of MT flattening caused by dynein-binding consistent with the flattening of B-tubule?

Our results suggest that DNAH7 prefers binding to local curvatures flatter than those seen in 13-protofilament microtubules (i.e. <27.7°). In the isolated axonemal doublet structure reported by (Ichikawa et al., 2017), the local curvature at the DNAH7 binding site (B-tubule protofilaments 9/10) is ~25°. This would be consistent with the B-tubule being optimized for binding however, it would be interesting to see how this curvature manifests in the context of the axoneme with its full array of binding partners and as it bends.

We now make note of this in the Discussion section, saying “We note that the inner proteins maintain the local curvature of the inner arm dynein binding site at ~25°, which would favour DNAH7 binding (Ichikawa et al., 2017).”

Reviewer #3:

[…] Overall, I think this is an outstanding paper, of very high interest in both biological insight and technological advance. I am not qualified to review the details of their cryo-EM algorithms, so I will take the author's word for the quality of the new technique. I find the biological insights to be stimulating, and think they open up very interesting new questions about the role of distinct axonemal dynein classes in flagellar movement.The only concern I have is that the MT distortion is observed on reconstituted MTs made from brain tubulin, an unnatural lattice for the axonemal MTBD. Since the natural lattice for this domain is the B1 doublet, how can we be sure the distortion observed is not an artifact of using single MTs? The authors should probably mention this caveat. Additionally, the authors make a fairly convincing argument that the flap binding to a neighboring protofilament is what may cause the lattice distortion. A formal test of this hypothesis would make the manuscript stronger. Mutation or deletion of the flap region could be done to examine if this restores the MT lattice to its original shape.

We agree with the reviewer’s comments that the differences between the microtubule doublet and in vitro polymerized microtubules means that the in vivo effect of DNAH7 is potentially different to what is observed in our structure. To reflect this, we have added the sentence to our Discussion section:

“Conversely, as our study was performed on in vitro polymerized microtubules, we cannot rule out that DNAH7 binding has a different effect on axonemal doublets, where for example inner proteins may increase the microtubule stiffness (Ichikawa et al., 2017, 2019).”